# Robust deep learning object recognition models rely on low frequency information in natural images

Zhe Li[1◔]*, Josue Ortega Caro[1◔], Evgenia Rusak[2], Wieland Brendel[2], Matthias Bethge[2], Fabio Anselmi[1], Ankit B. Patel[1,3,4‡], Andreas S. Tolias[1,3,4‡]*, Xaq Pitkow[1,3,4‡]*

1 Department of Neuroscience, Baylor College of Medicine, Houston, Texas, United States of America, 2 University of Tübingen, Germany, 3 Department of Electrical and Computer Engineering, Rice University, Houston, Texas, United States of America, 4 Center for Neuroscience and Artificial Intelligence, Baylor College of Medicine, Houston, Texas, United States of America

◔ These authors contributed equally to this work.
‡ABP, AST, and XP also contributed equally to this work.
* zhel@bcm.edu (ZL); astolias@bcm.edu (AST); xaq@rice.edu (XP)

**Data Availability Statement:** All data and computational codes are released in the public repository https://github.com/lizhe07/blur-net.

## Abstract

Machine learning models have difficulty generalizing to data outside of the distribution they were trained on. In particular, vision models are usually vulnerable to adversarial attacks or common corruptions, to which the human visual system is robust. Recent studies have found that regularizing machine learning models to favor brain-like representations can improve model robustness, but it is unclear why. We hypothesize that the increased model robustness is partly due to the low spatial frequency preference inherited from the neural representation. We tested this simple hypothesis with several frequency-oriented analyses, including the design and use of hybrid images to probe model frequency sensitivity directly. We also examined many other publicly available robust models that were trained on adversarial images or with data augmentation, and found that all these robust models showed a greater preference to low spatial frequency information. We show that preprocessing by blurring can serve as a defense mechanism against both adversarial attacks and common corruptions, further confirming our hypothesis and demonstrating the utility of low spatial frequency information in robust object recognition.

## Author summary

Though artificial intelligence has achieved high performance on various vision tasks, its ability to generalize to out-of-distribution data is limited. Most remarkably, machine learning models are extremely sensitive to input perturbations such as adversarial attacks and common corruptions. Previous studies have observed that imposing an inductive bias towards brain-like representations can improve the robustness of models, but the reasons underlying this benefit were left unknown. In this work, we propose and test the hypothesis that the robustness of brain-like models can be accounted for by a low frequency

**Funding:** This work is supported by the Intelligence Advanced Research Projects Activity (IARPA) via Department of Interior/Interior Business Center (DoI/IBC) (D16PC00003 to AST and XP), National Eye Institute of the National Institutes of Health (R01EY026927 to AST), NEI/NIH Core Grant for Vision Research (EY-002520-37) and NeuroNex (1707400 to XP and AST). The funders had no role in study design, data collection and analysis, decision to publish, or preparation of the manuscript.

**Competing interests:** The authors have declared that no competing interests exist.

feature preference inherited from the neural representation. We designed a novel machine learning task to probe the frequency bias of different models and observed a strong correlation between that and model robustness. We believe this work serves as a first step towards understanding why biological visual systems generalize well to out-of-distribution data and provides an explanation for the robustness of state-of-the-art machine learning models trained with various methods. It also opens the door to applying computational principles of the brain in artificial intelligence, hence helping to overcome the fundamental difficulties faced by current AI methods.

## Introduction

Currently, deep neural networks are the state-of-the-art models for numerous computer vision tasks such as object detection [1], image recognition [2], semantic segmentation [3], etc. However, these models are often not robust, as demonstrated by their inability to generalize to new data distributions. For instance, current neural networks are not able to generalize to common corruption noise such as ImageNet-C [4], where network performance is stress tested against 15 different kinds of image corruption applied to the ImageNet dataset. Furthermore, these models seem to be extremely sensitive to targeted noise such as adversarial attacks [5]. In contrast, the human visual system does not seem to suffer from such problems: in particular, recognizing object identity is little affected by the common corruptions [6], and adversarial perturbations that break machine learning models are imperceptible to humans [5]. This difference in behavior between the brains and deep learning algorithms might be explained by differences in inductive bias, i.e. they learn different features from data. Accordingly, natural vision has an inductive bias, a bias towards which fixed network are learned by the optimization algorithm from a class of models given the set of training data, towards robust features, which means insensitivity to perturbations that do not change the perceptual relevant latent variables such as object identities. How can we instill the brain's inductive bias towards robust to targeted and random noise distortions to these deep learning algorithms? Recent work has shown that machine learning models that are encouraged to learn brain-like representations, a paradigm known as neural regularization, are also more robust to certain common corruptions such as Gaussian noise and adversarial attacks [7, 8]. Furthermore, other work has shown that models that explain more variation in primate primary visual cortical (V1) responses also tend to be more robust to adversarial attacks [9].

Much recent parallel work has also attempted to produce models that are robust to common corruptions (ANT [10], SIN [11], DeepAugment, AugMix [4]) and to adversarial attacks (PGD Training [12], TRADES [13]). These models achieve significant improvements upon baseline models by employing several other methods, including data augmentation, adversarial training, and anti-aliasing [14]. AutoAugment, a data augmentation method for common corruption robustness, has produced improvements mainly for high frequency common corruption noises [15]. Consistent with this work, we hypothesize that one of the reasons for the success of current robustness methods for both common corruption and adversarial attacks can be explained by a simple computational principle: models that are biased to rely on low spatial frequency information for object recognition are more robust. Here, we tested this hypothesis with several frequency-oriented analyses. Finally, we introduced a simple preprocessing step based on these principles that produced robust models with comparable performance to these more sophisticated methods.

## Results

### Neural regularization boosts model robustness

There are many ways to bias a machine learning model toward brain-like computations, such as architecture, learning rules, the training dataset or task's objective functions [16]. Among them, methods that use auxiliary loss functions to bias models towards brain-like representations are called neural regularization. We will focus on two particular forms of neural regularization: neural similarity regularization [7, 17] and neural response regularization [8]. Both methods directly encourage a model to learn a brain-like representation of the visual stimuli.

The neural responses of natural images are located on a low dimension manifold in the response space [18]. Similarly, for any machine learning model, features of images stimuli also form their own manifold embedded in model feature space. Neural similarity regularization adjusts the manifold in the model's hidden layers so that it is biased towards the same geometry as the manifold in the neural response space (Fig 1), while performing benchmark tasks like image classification at the same time. Li et al. [7] showed that a ResNet18 model regularized to incentivize its representational similarity [19], which is the pair-wise cosine similarity between the representation for a set of images, to be more like the one measured in mouse V1 is more robust against Gaussian noise and adversarial attacks. Another way to induce brain-like representations is to request the model to predict neural responses of the visual input as an auxiliary task. Safarani et al. [8] showed that a VGG19 model co-trained with monkey V1 data, where the VGG19's core was used to predict neural responses directly, is able to improve robustness against common corruptions on Tiny ImageNet. Since the neural readout module is shallow, the VGG19 core has to encode neural features to make the co-training work.

In this study, we expanded both neural regularization, either using mouse's representational similarity matrix or using the monkey responses directly, by testing model robustness against various common corruptions and adversarial attacks.

Since neurally regularized models are trained on grayscale images, we used grayscale versions of the CIFAR10-C and TinyImageNet-C datasets for evaluation. Models regularized with either mouse or monkey V1 representations have higher accuracy at different severity levels of common corruptions compared to baseline models (Fig 2a and 2e), though their performance on clean images is slightly worse. Targeted gradient-based boundary attacks [20] were performed to find the minimum perturbations needed to change model predictions to wrong categories. The use of strong attacks is critical in evaluating adversarial robustness, as weak attacks such as FGSM only provide a loose upper bound on the minimum adversarial

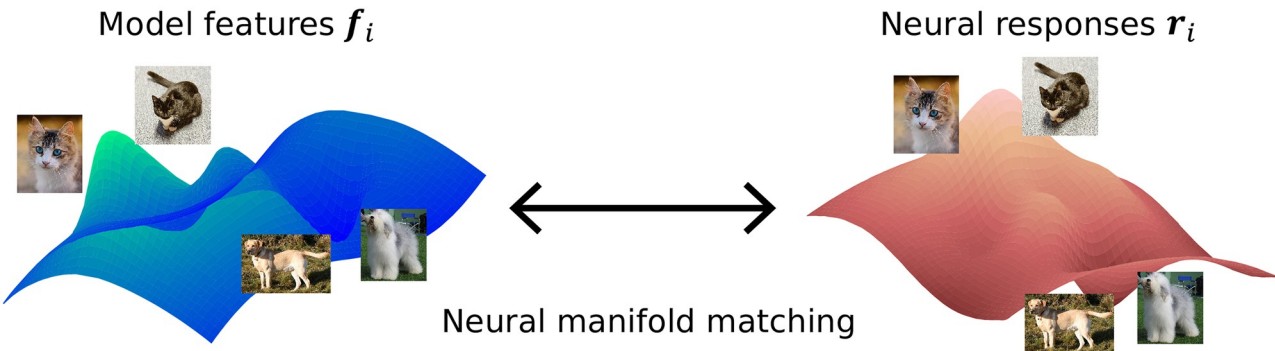

**Fig 1. Schematic of neural similarity regularization.** Machine learning models are trained so that the stimuli manifold in the model feature space resembles the manifold in the neural response space. The most simple version considers only pairwise relationships among stimuli instances. Two images that are close in the neural response space should also be close in the model feature space.

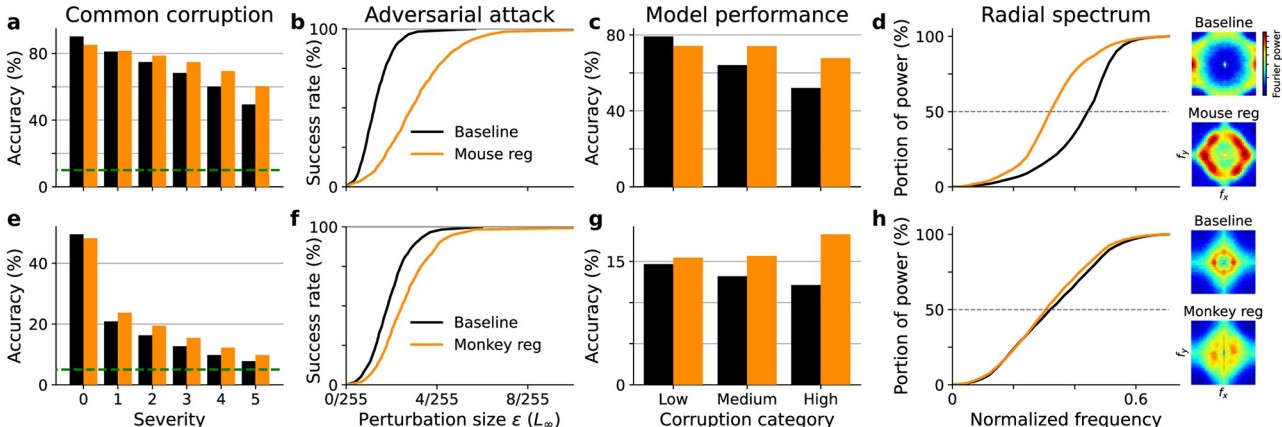

**Fig 2. Neural regularization boosts model robustness and makes it less sensitive to high frequency component of the input.** A ResNet18 model (orange in a-d) was trained for grayscale CIFAR10 with mouse neural similarity regularization [7], a VGG19 model (orange in e-h) was trained for grayscale TinyImageNet with monkey neural response regularization [8]. (a) Grayscale CIFAR10 classification accuracy against common corruptions at different severity levels. Average accuracy over all corruptions are reported for a baseline ResNet model (black) and a mouse regularized model (orange). (b) Success rate of targeted attacks at different perturbation budget $\epsilon$, using the boundary attack [20] with an $L_\infty$ metric. (c) Classification accuracy against different types of corruptions, broken down into three groups based on their frequency characteristics (Table A in S1 Appendix). Model performance is averaged over all severity levels. (d) Radial profile of the Fourier spectrum of adversarial perturbations. We found the minimal adversarial perturbations of all testing images, and calculated the averaged Fourier spectrum thereof, where blue is minimum and red is maximum values of each heat map respectively (insets), a logarithm scale color map is used for better visualization. The portion of power under different frequency thresholds are compared between baseline and neurally regularized models. The abscissa is the absolute value of the spatial frequency, normalized by sampling frequency $f_s$. (e–h) Same as a–d, except comparing a baseline VGG model with a model co-trained with monkey neural data on the grayscale TinyImageNet dataset [21].

perturbation size. We verified the effectiveness of boundary attacks by extensive comparison between many off-the-shelf methods, and are confident it characterizes the adversarial robustness of models. Evaluation on the testing set showed that neurally regularized models need larger adversarial perturbations on average (Fig 2b and 2f), i.e. they are more robust against adversarial attacks. The attack size that gives 50% success rate is $\epsilon = 1.25/255$ for the baseline ResNet and $\epsilon = 2.89/255$ for the mouse regularized one. If we look at the distribution of minimum perturbation size needed for each image, the mean and standard deviation is $\epsilon = (1.34 \pm 0.70)/255$ for the baseline ResNet, and $\epsilon = (3.09 \pm 1.61)/255$ for the mouse regularized one. Similarly, the attack size that gives 50% success rate is $\epsilon = 1.84/255$ for the baseline VGG and $\epsilon = 2.46/255$ for the monkey regularized one. And the minimum perturbation size for all images is $\epsilon = (1.93 \pm 0.89)/255$ for the baseline VGG, and $\epsilon = (2.64 \pm 1.26)/255$ for the monkey regularized one. In the following text, we use the mean value of minimum perturbation sizes to characterize adversarial robustness of the model.

A natural question is: what does the neural regularization do? First, we decided to explore the structure of the neural similarity matrix of mouse V1 responses with a simple decomposition analysis. We calculated the eigenvalues and eigenvectors of the similarity matrix and computed the linear approximation of each principal component with respect to the image. We observed that geometry of mouse V1 representation is well approximated by a small number of principal components, and the spatial tuning of them reveals low frequency structure (Fig D in S1 Appendix). This has led us to ask if the bias towards low frequency features is inherited by neurally regularized models. We divided the 15 common corruptions in the CIFAR10-C and TinyImageNet-C datasets [22] into three categories based on their frequency spectra (Table A in S1 Appendix). Low-frequency corruptions: 'snow', 'frost', 'fog', 'brightness', 'contrast'; medium-frequency corruptions: 'motion_blur', 'zoom_blur', 'defocus_blur', 'glass_blur', 'elastic_transform', 'jpeg_compression', 'pixelate';

high-frequency corruptions: 'gaussian_noise', 'shot_noise', 'impulse_noise'. Fourier spectra of different corruptions are shown in Fig A in S1 Appendix. The biggest performance boost in model classification accuracy comes from the category with the high-frequency corruptions (Fig 2c and 2g).

We also compared the adversarial perturbations for baseline models and neurally regularized ones. We performed a Fourier analysis on the minimal adversarial perturbation found through our boundary attack [20], and calculated the average frequency spectrum for different models (insets of Fig 2d and 2h). We observed that the mouse and monkey neurally regularized models contain relatively higher low-frequency components than the baseline model. To quantify this frequency shift, we characterized the frequency preference by a radial profile of the spectrum of the adversarial perturbation, *i.e.* the power of all Fourier components whose frequency is smaller than certain values. We found the radial profile of the adversarial perturbation spectrum for the neurally regularized model is shifted toward lower frequencies (Fig 2d and 2h). Furthermore, we can quantify this shift by the half power frequency $f_{0.5}$, which is the frequency where half of the Fourier power lies below (marked by the dashed line in Fig 2d and 2h). $f_{0.5} = 0.316 \pm 0.015$ for mouse regularized ResNet and $f_{0.5} = 0.442 \pm 0.009$ for baseline, with mean and standard deviation estimated from 1000 images (Fig 2d). Similarly, $f_{0.5} = 0.306 \pm 0.020$ for monkey regularized VGG and $f_{0.5} = 0.316 \pm 0.034$ for baseline (Fig 2h). The half power frequency $f_{0.5}$ is smaller for the neurally regularized model compared with baseline, though effect on the monkey regularized model is much weaker than the mouse regularized one.

## Hybrid image experiment

To directly probe the frequency bias of models, we next designed a new dataset of hybrid images constructed by mixing the low-frequency components and high-frequency components from two different images [23]. We select two images belonging to two different categories, and examine whether the model prediction on the mixed image is consistent with either component. For a given mixing frequency $f_{\mathrm{mix}}$, we combine the Fourier components of an image whose frequencies are smaller than $f_{\mathrm{mix}}$ with Fourier components of another image whose frequencies are larger than $f_{\mathrm{mix}}$, and then use an inverse Fourier transformation to get a hybrid image (Fig 3a, see Methods for more details).

We denote the probability that a hybrid image is classified as the low-/high-frequency seed image class as $p_{\mathrm{low}}$ and $p_{\mathrm{high}}$ respectively, and calculate the probability difference $p_{\mathrm{low}} - p_{\mathrm{high}}$ at different mixing frequency values. When the mixing frequency is very small, the hybrid image is close to the high frequency seed image, therefore $p_{\mathrm{high}}$ is high for a properly trained model. Likewise, when the mixing frequency is big, $p_{\mathrm{low}}$ is high. We are interested in the mixing frequency when $p_{\mathrm{low}} = p_{\mathrm{high}}$, as it characterizes the frequency bias of a model. We term this frequency the reversal frequency $f_{\mathrm{rev}}$. The result shows that $f_{\mathrm{rev}}$ is smaller for a mouse regularized model ($0.371 \pm 0.0006$, mean $\pm$ standard deviation estimated from 4 randomly permuted hybrid image datasets) than a baseline model ($0.528 \pm 0.0009$), indicating that the neurally regularized model is more likely to classify the hybrid image as the class of the low-frequency component (Fig 3b). This provides strong evidence that the reason behind the robustness gain by mouse neural regularization could be a bias towards low-frequency features. The same experiments and analysis were done for monkey regularized models as well, but the low-frequency bias is smaller compared to mice. $f_{\mathrm{rev}}$ is $0.376 \pm 0.003$ and $0.426 \pm 0.002$ for monkey regularized VGG19 and a baseline model, respectively (Fig B in S1 Appendix).

Since it appears the neural induced robustness can be largely accounted for by effectively changing the sensitivity to different spatial frequency, it is straightforward to implement such

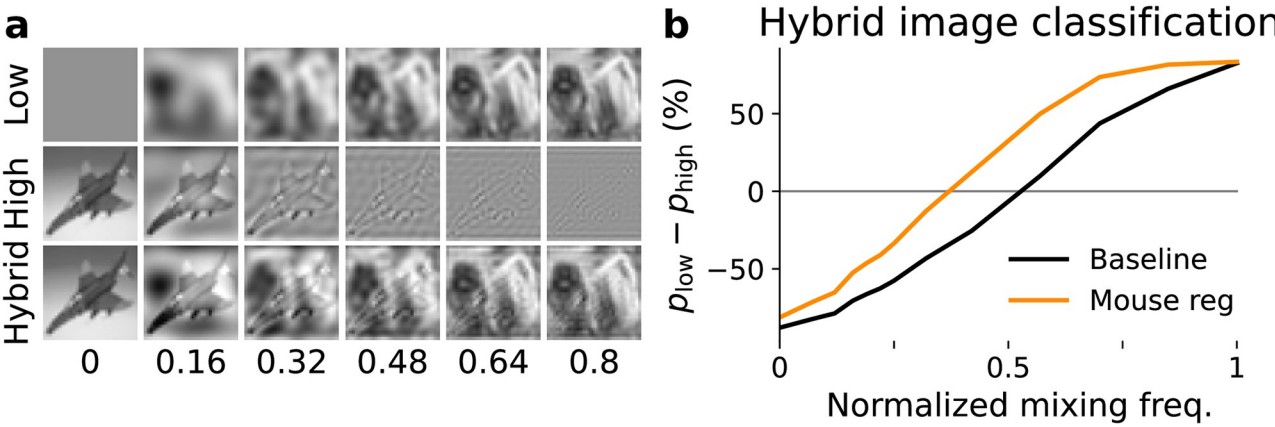

**Fig 3. Probing frequency sensitivity of mouse regularized model using hybrid images.** (a) Examples of hybrid images at different mixing frequencies. Hybrid images are constructed by mixing the low-frequency component of one image and the high-frequency component of another, while the two seed images belong to different categories. The range of mixing frequency's values are normalized by the Nyquist frequency. (b) Model predictions on hybrid images at different mixing frequencies. As more low-frequencies from one image are included, the probability that a network reports its label $p_{\text{low}}$ increases. The reversal frequency $f_{\text{rev}}$ where $p_{\text{low}} = p_{\text{high}}$ is smaller for the mouse regularized model ('neural') than for the baseline model ('base').

principle via a simple preprocessing of images. We propose two simple methods for filtering images and will describe them in the next section.

## Frequency analysis on robust models

We then asked if other models that are not regularized with neural data, but engineered to be robust to common corruptions and adversarial attacks also have this low frequency bias. To this end, we analyzed and compared several robust models trained on CIFAR10, most of which were downloaded from RobustBench [24]. Some of the models are trained for adversarial robustness and some are trained for common corruption robustness. A full description of models is listed in Table B in S1 Appendix. Since the mouse regularized model in Fig 2 was trained on grayscale CIFAR10, it is not included in this comparison.

Two additional models were trained and included in this comparison. Both models were constructed by attaching a preprocessing layer before a ResNet18 model. The parameters of the preprocessing are not trainable but fixed beforehand. One is called the 'blur' model as the preprocessing is a convolution with a Gaussian kernel of standard deviation $\sigma$. The other is called the 'PCA' model as the preprocessing keeps only the first $K$ principal components (PCs) calculated from all training images. $\sigma$ and $K$ are chosen such that the classification accuracy on CIFAR10 of both models are around 90%, similar to other robust models. More details are included in the Methods section.

The 'blur' model directly biases the model towards low-frequency features, because the high frequency features are attenuated during the preprocessing step. The filtering of features based on their spatial frequency can be treated as a special form of reweighting of input principal components, since principal component analysis on natural images with translation invariance recovers the Fourier basis, and the variance explained by a feature usually decreases monotonically as frequency increases. Hence a 'PCA' model that explicitly projects the original image onto a low dimensional space spanned by the first PCs is also included in the comparison. The hypothesis behind 'PCA' model is that the directions parallel to the data manifold are robust features, and perturbations that are orthogonal to the manifold can be safely removed. Since natural images have the greatest variance in low frequencies, the distance of any data point to a

class boundary are longest along low frequency dimensions, and thus a model that mainly uses these low frequencies should be the more robust. Previous work also found that this kind of PCA data preprocessing increases robustness to adversarial attacks [25].

Similar to the analysis done on neurally regularized models, we calculated the Fourier spectrum of minimum adversarial perturbations for all robust models. A fixed set of 1000 images were selected from testing set, and the incorrect class targets for each image was also fixed. The adversarial perturbations for robust models, including the ones not specifically trained for adversarial robustness, contained more low spatial frequency components, while in the baseline models, adversarial perturbations were dominated by high spatial frequency components (Fig 4a). To better visualize the differences of adversarial spectrum shape, we plotted the radial

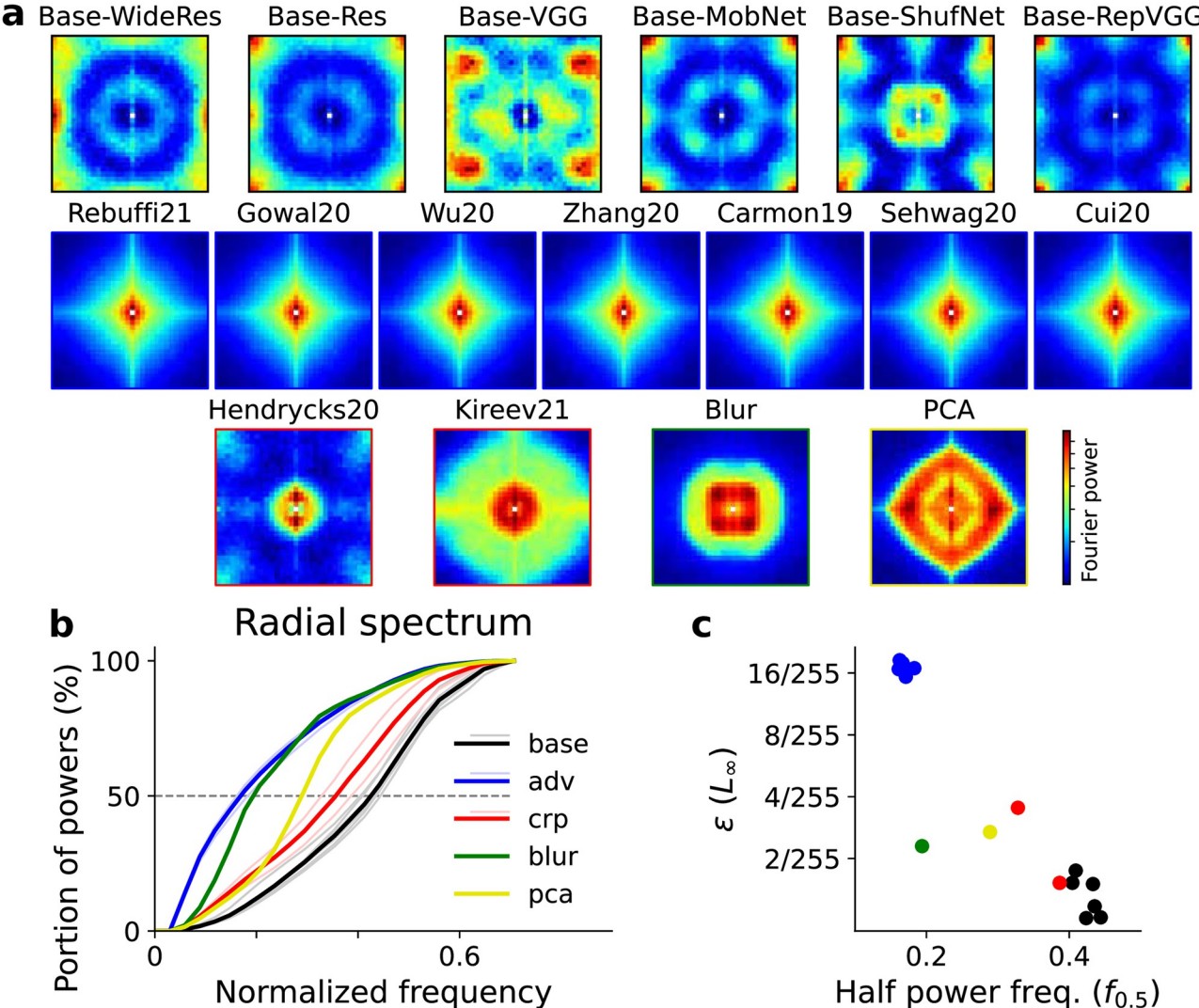

**Fig 4. Frequency analysis of adversarial attacks on robust models trained on CIFAR10.** (a) The Fourier spectrum of the minimal adversarial perturbations of different models, including six baseline models ('base'), seven models trained for adversarial robustness ('adv'), two models for corruption robustness ('crp'), one model with preprocessing by blurring ('blur'), and one with preprocessing by PCA compression ('pca'). Model details are listed in Table B in S1 Appendix. The spectrum is averaged over 1000 images, and color maps are normalized separately for each panel. (b) Radial profiles of adversarial perturbation spectra. Light thin lines represent each individual model, while thick lines are the average within each group. The frequency where each line crosses 50% is denoted as half power frequency $f_{0.5}$. (c) Scatter plot of minimum adversarial perturbation size versus $f_{0.5}$ for all models.

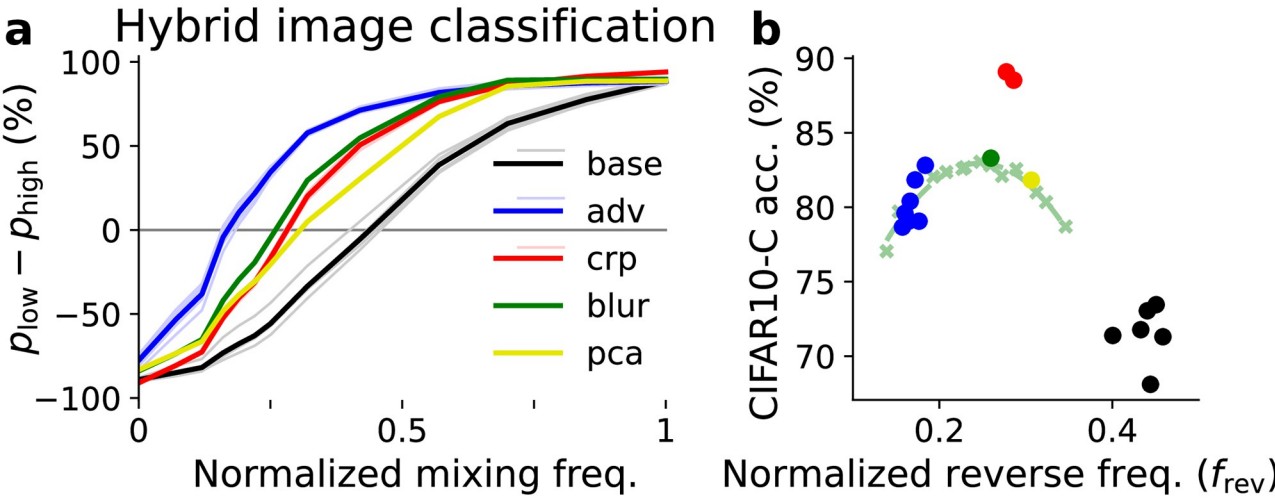

**Fig 5. Hybrid CIFAR10 image classification performance of robust models.** (a) Difference between probability of choosing the low frequency label vs the high frequency label of hybrid images. As in Fig 4, light thin lines represent each individual model and thick lines are the average within each group. (b) Scatter plot of model accuracy on CIFAR10-C dataset versus reversal frequency $f_{rev}$ in hybrid image classification. The dashed green line is the performance of a series of 'blur' models, using different degree of low-pass filtering. The left end corresponds to $\sigma = 3$ pixels and the right end corresponds to $\sigma = 1$ pixel, while the green dot is the model with $\sigma = 1.5$ pixel listed in Table B in S1 Appendix.

spectrum for each model. The portions of the spectral power within different spatial frequencies are compared (Fig 4b), and the half power frequency $f_{0.5}$ is marked by the dashed line. Compared with the baseline model, all robust models have smaller $f_{0.5}$, indicating the minimum adversarial perturbations for them have lower frequencies. The size of average minimal perturbations are plotted against $f_{0.5}$ in Fig 4c, revealing a negative correlation between adversarial robustness and the frequency bias of the model.

We next tested model predictions on hybrid images. Hybrid images for RGB CIFAR10 are constructed similarly as in Fig 3a. The reversal frequencies $f_{rev}$ for all models are plotted against the classification accuracy on the CIFAR10-C dataset (averaged over all corruptions and severity levels) in Fig 5b. The result shows that all models are more robust to common corruptions than the baseline, even those trained for adversarial robustness. They all have a low frequency bias characterized by $f_{rev}$, which suggests that low frequency information is more important to these models. However, one interesting observation is that compared to adversarial attacks, common corruption robustness seems to cap in an specific $f_{rev}$. For example the adversarially robust models (blue) have smaller corruption accuracy compared to the other robust models. This seems to indicate a trade-off, where using too few frequencies ends up affecting performance. We decided to explore this by using different 'blur' models with increasing blurring kernels' size. In Fig 5b, we observe that most models except the ones trained for common corruption robustness lie close to the curve (light green) fit to 'blur' models (light green crosses). This is an indication that the decrease in performance as a function of $f_{rev}$ can be partly explained through the frequency preference of the models.

Next we explored this relationship between robustness and frequency bias on models trained for ImageNet classification. In addition to a baseline model, we analyzed two adversarially trained models, six models trained for robustness to common corruption and a model with blurring preprocessing layer (Table C in S1 Appendix). In Fig 6a, we plot the minimum adversarial perturbation size $\epsilon$ with respect to the half-power frequencies, $f_{0.5}$, of adversarial perturbation spectra. We can observe that the minimum perturbation size is negatively

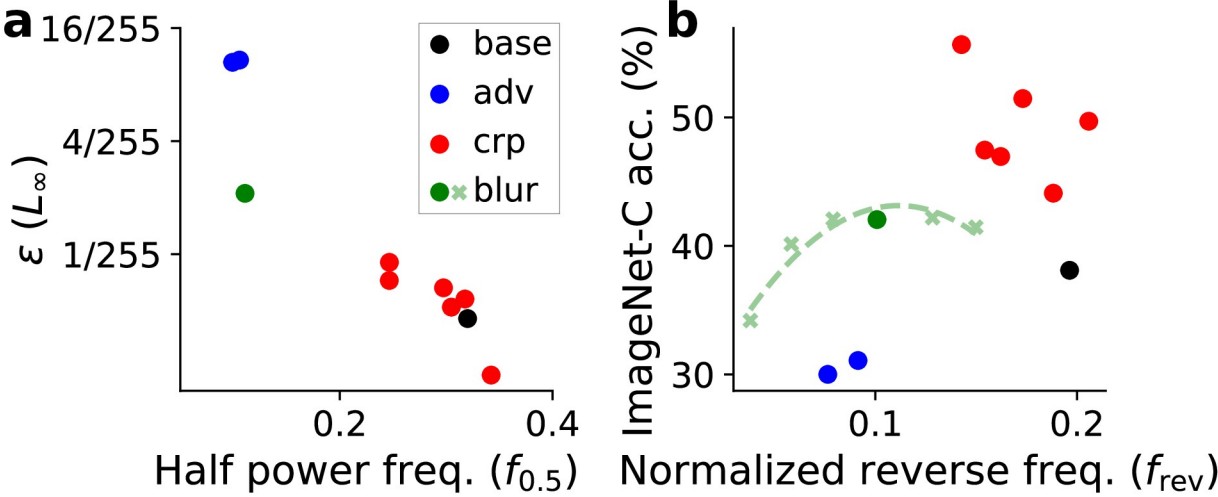

**Fig 6. Frequency analysis of models trained on ImageNet.** One baseline model ('base'), two models ('adv') trained for adversarial robustness, and six models ('crp') trained for corruption robustness are compared. (a) Minimum adversarial perturbation size $\epsilon$ versus the half-power frequency $f_{0.5}$ calculated from adversarial perturbation spectra. (b) Model accuracy on ImageNet-C dataset versus reverse frequency $f_{rev}$ calculated from hybrid image experiment.

correlated with half-power frequencies, this indicates that robustness to adversarial attacks is correlated with adversarial attacks that have more energy in low frequencies.

Furthermore, Fig 6b shows corruption accuracy versus the reverse frequency, $f_{rev}$, for the ImageNet-C dataset. Here we observe that the common corruption robust models are more robust than the baseline and have smaller $f_{rev}$, consistent with our findings in CIFAR10 models. However the adversarially robustness models are less robust than baseline even though they have smaller $f_{rev}$. This is probably because the clean performance of adversarially robust models on this dataset is too low, therefore they are not comparable to baseline (Table C in S1 Appendix).

Similar to the analysis done for CIFAR10 models, we trained a series of 'blur' models with different blurring $\sigma$ (light green crosses in Fig 6b). We observe the same 'Goldilocks' behavior in Fig 6b, that either too much or too little blurring decreases model performance on corrupted images. Other models deviate from this 'blur' model manifold more compared with the results for CIFAR10 dataset (Fig 5b), suggesting that a simple low-/high-frequency view on the robustness is less accurate for models trained on larger images. For example, perhaps some robust features in such dataset are not entirely low-frequency. While 'blur' models for CIFAR10 are trained from scratch, 'blur' models for ImageNet are initialized by a pre-trained baseline ResNet50 model ('base' in Fig 6) and fine-tuned after the fixed blurring layer is added. This might induces some bias on the 'blur' models towards the baseline model and affects the frequency tuning properties.

## Discussion

Recent studies have linked brain-like representations to robustness of deep network models [7–9]. By introducing a novel hybrid image task, we were able to study the frequency preference of a neural networks, and show that these brain-like models were more robust mainly through a preference towards low spatial frequency features. This work follows in spirit the approach of Geirhos et al. [11], where the authors combined the shape of an input and the texture of another one to produce an input that had features with "conflicting information"

towards different classes. This approach of producing cue-conflict style tasks seems to be useful for understanding the model preference towards certain features and not others.

Furthermore, we have shown that this bias is present in an extensive group of machine learning models trained for robustness: we found that robustness to common corruptions and adversarial examples are correlated with this low frequency bias. However, there seems to be a difference between robust models on CIFAR10 vs ImageNet. Adversarially robust models are more robust to common corruptions on CIFAR10 but not on ImageNet. Previous work has also shown that adversarially robust models are not robust to common corruptions [26]. However, they attributed this behavior to both perturbations having qualitative different properties. In our case, we argue that adversarial robustness in ImageNet dataset is more difficult to achieve, and therefore the models we analyzed cannot handle perturbations as large as the ones in common corruptions compared to CIFAR10 trained models. All of this seem to indicate that robustness to different datasets might be achieved through different methods but low-frequency preference seem to be a key component of current robust models.

The mouse regularized model shows a clear bias towards low spatial frequency compared with the baseline (Figs 2d and 3b), which must be inherited from the neural representation of mouse V1. We hence decomposed the mouse V1 manifold and looked at the spatial tuning of major components. We found that the dominant dimensions of neural population responses can be approximately described by linear receptive fields with low spatial frequency (Fig D in S1 Appendix), and that is the origin of low frequency preference of mouse regularized model. Although we do observe statistically significant effect of low frequency bias in monkey regularized model, it is much weaker than that in the mouse regularized one. We speculate it is because monkey V1 representation encodes other robust features that are not low spatial frequency. Analysis in Safarani et al. [8] shows monkey regularized model emphasizes the salient regions of images more than other control models, which suggests that the robustness is due to more than just a change on frequency sensitivity.

Previous physiological experiments [27–29] have measured the tuning properties of neurons in mouse and monkey visual cortices. Here we take the values from Table 1 of Van den Bergh et al. [29] regarding the preferred spatial frequency of a neuron, which are 0.04 cycles/deg for mouse V1 and 2.43 cycles/deg for monkey V1. We can safely assume each natural image used in the neural regularization studies [7, 8] contains one object, and treat the image size as a universal length unit. In the mouse regularization work, each image used in the experiment is approximately 67.5 visual degrees in height, therefore the preferred frequency of V1 neuron is about 2.7 cycles per image. In the monkey regularization work, each image is approximately 6.7 visual degrees, hence the preferred frequency of V1 neuron is about 16.3 cycles per image. The difference in normalized preferred spatial frequencies is also consistent with the observation that mouse regularized model shows more low frequency bias and robustness than the monkey regularized counterpart.

In addition, we were able to use the strategy of biasing towards low spatial frequencies and enforced it with a simple preprocessing layer to produce robustness on par with this group of robust machine learning methods. Previous work has also tried to use frequency information to produce robust models for ImageNet [14, 30, 31]. For example, antialiasing has been shown to improve robustness to common corruptions when introduced before or after the nonlinearities of different models [14, 31]. However, this method seems to behave differently than our preprocessing models. For example, these anti-aliasing models decrease in performance as you go into the high frequency corruptions, similar to the baseline models. This is in contrast to the other robust models that perform better on high frequency corruptions compared to low frequency ones. As the authors explained in their work, the anti-aliasing methods focus more on producing shift-invariant models while preserving as much high frequency information as

possible. This might be the reason why this model behaves differently than the 'blur' or 'PCA' models which explicitly attenuate high frequency information from the input, and why our models behave more similarly to neurally regularized and data augmented models. This indicates that there can be different approaches to producing robust models to common corruptions with focus on different properties of the frequency spectrum.

However, as previous work has shown, using low frequency bias has its limitations. For example, Yin et al. [15] have shown that training a model on the "Fog" common corruption only using the frequency, but not the spatial information is insufficient to generalize to the same "Fog" common corruption during testing. This makes sense given that the corruption has a very specific spatial structure that does not depend only on the frequency. In addition, as we observed in Figs 5b and 6b, the bias towards low frequencies can be too strong, such that the performance deteriorates. This suggests that to reduce the gap between humans and machine learning models in out-of-distribution generalization, we must move beyond this low frequency-based preference found in current robust machine learning models and find a better and more principled inductive bias.

One possibility to achieve this grand goal is to use ideas from neuroscience. As our and previous work has suggested, neuroscience has provided inspiration to machine learning, but specific paradigms to directly translate biological scientific insights and data into a performance improvement in machine learning are largely absent. The neural regularization approach is a demonstration that this direction can help engineer more intelligent algorithms to usher the new field of NeuroAI. Furthermore, given that current machine learning models are still not robust to adversarial attacks and common corruptions, this work is just the start of bridging the gap between natural and artificial intelligence.

Besides making the model representation be more brain-like, recent research has also introduced other neural features into machine learning models to make it more robust, for example using biologically-constrained Gabor filters as the first layer and adding stochasticity to mimic the spiking responses in neural systems [9]. The more biological Gabor filters functions much like our blurring preprocessing layer, since they both gains Fourier components to enhance features in certain spatial frequency interval. We do not put too much effort on the the stochasticity as there have been evidence that such defense mechanism usually cause gradient masking as obfuscated gradients [32] and overestimated the model robustness. When properly attacked, for example computing the expectation over instantiations of randomness, the appeared adversarial robustness is gone [32]. Evaluation of stochastic models does not follow the same protocol as that of deterministic ones [24], we therefore focus only deterministic models as they gain the robustness through using different representation from that in baseline models.

One recent research [33] measured the preferred spatial frequency of individual model units via in-silico electrophysiology experiments, and showed that representation of robust models are more aligned with macaque V1 neurons in terms of the distribution of preferred spatial frequencies compared with non-robust models. Based on the eigenspectrum analysis, they suggested that the robustness is due to smaller portion of high-frequency tuning units in the robust models. Our work is complementary to their results as we analyze the model frequency preference on the behavioral aspect instead of individual model units.

## Methods

### Hybrid images

We randomly select two images $I_{low}$ and $I_{high}$ from two different categories as seed images for low-frequency and high-frequency components respectively. The 2D Fourier transform of these two images are denoted as $\tilde{I}_{low}$ and $\tilde{I}_{high}$, and we define a binary mask on frequency

domain as

$$M(\boldsymbol{f}) = \begin{cases} 1, & \|\boldsymbol{f}\| \leq f_{\text{mix}} \\ 0. & \text{otherwise} \end{cases} \tag{1}$$

The Fourier transform of the hybrid image is thus the combination of $\tilde{I}_{\text{low}}$ and $\tilde{I}_{\text{high}}$ through $M$, according to $\tilde{I}_{\text{hybrid}} = \tilde{I}_{\text{low}} \odot M + \tilde{I}_{\text{high}} \odot (1 - M)$, where $\odot$ denotes an element-wise product. The hybrid image $I_{\text{hybrid}}$ is calculated from $\tilde{I}_{\text{hybrid}}$ using an inverse Fourier transform.

## Blur model

We designed a blur model to explicitly implement a frequency bias. Our blur model is simply a ResNet model prepended with a blurring layer. The blurring layer is a linear convolutional layer whose kernel weight is fixed as

$$w(\Delta x, \Delta y) = \frac{1}{Z} \exp\left(-\frac{\Delta x^2 + \Delta y^2}{2\sigma^2}\right) \tag{2}$$

in which $Z$ is the normalization factor. The blurring layer can pass a gradient back to the input image, so gradient-based adversarial attacks can be performed. Clean performance of the 'blur' model decreases as $\sigma$ increases because more information is discarded. We choose the value of $\sigma$ such that the trained model has a similar classification accuracy on CIFAR10 testing set compared with other robust methods; $\sigma = 1.5$ pixel is selected.

## PCA model

Similar to the blurring model, the PCA model is also a ResNet model with a non-trainable pre-processing layer. Treating the input image as a vector $\boldsymbol{x}$ whose dimension $N$ is the number of pixels, we first performed principal component analysis (PCA) on the training set and obtained the eigenvectors $\boldsymbol{v}_i$ $(1 \leq i \leq N)$ sorted by their eigenvalues. We took the first $K$ eigenvectors, denoted by a $K \times N$ matrix $\boldsymbol{W}_K$, and constructed the filtering matrix

$$\boldsymbol{W} = \boldsymbol{W}_K \cdot \boldsymbol{W}_K^{\mathsf{T}}. \tag{3}$$

The PCA preprocessing layer is a linear layer whose input and output dimensions are both $N$, with its weights given by the matrix $\boldsymbol{W}$. This layer projects the original image onto the subspace spanned by the first $K$ eigenvectors, and effectively denoises the image. Similarly to the blur model, the PCA model allows gradient backward passes to the input image, and can be attacked with gradient-based adversarial attacks. The value of $K$ is chosen so that the trained model has a similar clean performance compared with other models, and $K = 512$ was selected for the CIFAR10 dataset.

## Model training and evaluation

The details about training mouse regularized models can be found in Li et al. [7]. The details about training monkey regularized models can be found in Safarani et al. [8]. Models for standard CIFAR10 and ImageNet tasks are fetched through RobustBench [24] except for the pre-processing ones.

To evaluate adversarial robustness of models, we used Foolbox [34, 35] to perform targeted attacks using both gradient-based boundary attacks and projected gradient descent attacks (PGD) on a fixed subset of testing images. For each image, the attack target class is predetermined in a random way. We gather all candidates proposed by different attack algorithms and

random seeds for each image, and pick the one with minimum perturbation size as the final adversarial example.

Codes for training 'blur' and 'PCA' models, robustness evaluation of all models can be found in https://github.com/lizhe07/blur-net.

## Disclaimer

The U.S. Government is authorized to reproduce and distribute reprints for Governmental purposes notwithstanding any copyright annotation thereon. The views and conclusions contained herein are those of the authors and should not be interpreted as necessarily representing the official policies or endorsements, either expressed or implied, of IARPA, DoI/IBC, or the U.S. Government.

## Supporting information

**S1 Appendix. Additional analysis and results.**
(PDF)

## Author Contributions

**Conceptualization:** Zhe Li, Josue Ortega Caro, Ankit B. Patel, Andreas S. Tolias, Xaq Pitkow.

**Data curation:** Zhe Li, Josue Ortega Caro.

**Formal analysis:** Zhe Li, Josue Ortega Caro.

**Funding acquisition:** Ankit B. Patel, Andreas S. Tolias, Xaq Pitkow.

**Resources:** Evgenia Rusak, Wieland Brendel.

**Supervision:** Ankit B. Patel, Andreas S. Tolias, Xaq Pitkow.

**Writing – original draft:** Zhe Li, Josue Ortega Caro, Ankit B. Patel, Andreas S. Tolias, Xaq Pitkow.

**Writing – review & editing:** Zhe Li, Josue Ortega Caro, Evgenia Rusak, Wieland Brendel, Matthias Bethge, Fabio Anselmi, Ankit B. Patel, Andreas S. Tolias, Xaq Pitkow.

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
