## [Decision Letter · Decision Letter 0]

28 Apr 2022

Dear Li,

Thank you very much for submitting your manuscript "Robust deep learning object recognition models rely on low frequency information in natural images" for consideration at PLOS Computational Biology.

As with all papers reviewed by the journal, your manuscript was reviewed by members of the editorial board and by several independent reviewers. In light of the reviews (below this email), we would like to invite the resubmission of a significantly-revised version that takes into account the reviewers' comments.

We cannot make any decision about publication until we have seen the revised manuscript and your response to the reviewers' comments. Your revised manuscript is also likely to be sent to reviewers for further evaluation.

Sincerely,

Xuexin Wei

Associate Editor

PLOS Computational Biology

Wolfgang Einhäuser

Deputy Editor

PLOS Computational Biology

Reviewer's Responses to Questions

**Comments to the Authors:**

**Please note that the reviews of reviewer 1 and 2 are available as pdf.**

Reviewer #1: This paper shows that the deep neural network models that are robust to adversarial images or data augmentations share one common feature of preferring the low frequency information in the natural images.

The authors started their analysis from the neural regularized models and found that the mouse-regularized model has a strong preference of the low frequency feature.

They then validated that the other publicly available robust models also share this preference and further proposed the blurring preprocessing as a defense strategy against the attacks.

At a high level, I think the main point of this article (robust networks prefer low-frequency features) is indeed supported by the presented evidence.

This result will be useful for the community to better understand these robust networks and then propose the networks that are even more robust.

However, I think there are several (possibly critical) issues that need to be addressed to make the whole story coherent.

Detailed comments are in the attachment.

Reviewer #2: Uploaded as an attachment

Reviewer #3: Li et al. test an interesting computational hypothesis. Namely that computational models of vision are more robust to adversarial perturbations and image corruptions if they have a bias towards lower spatial frequencies. The authors show via power spectral analysis of adversarial images and a clever ‘mixing’ experiment that neural regularized models (i.e., image classification models biased to have a similar representational geometry as mice/primate visual activity) have a bias towards lower spatial frequencies. In addition, they show that the neural regularization models are more robust by being less susceptible to ‘common corruptions’ (CIFAR10-C, ImageNet-C; Geirhos et al., 2016) and by yielding higher minimal perturbation distances for adversarial images generated under these models. The authors then suggest that a low frequency bias might be useful for robust object recognition in general. They study the spatial frequency bias of robustly trained models (adversarially trained or trained on ‘common corruptions’) and find that these indeed have a bias towards lower spatial frequencies compared to a baseline model. They also compare robust models to a model that includes a simple blur or PCA preprocessing step and show that some aspects of robustness can indeed be explained by biasing the model to lower spatial frequencies in the input.

I enjoyed reading the paper and I consider its contribution to be valuable for our understanding of model robustness. I do have several comments though where I think the paper should be improved.

Comments:

1) I really like the idea of the blur model in the second part of the paper. It is the most direct implementation of the computational hypothesis stated in the paper that a preference for lower spatial frequencies increases the robustness of a model. This model therefore has the highest explanatory value w.r.t the main claim of the paper. Unfortunately, the blur model is missing in the first part of the paper. It is therefore not clear whether the neural regularization model inherits the lower spatial frequency bias *and* the robustness of the mice/primate visual system or whether the neural regularization model is more robust *because* it inherits the lower spatial frequency bias (which is the computational hypothesis of the paper). I would suggest to include blur models in the first part of the paper to tease these possibilities apart.

2) The evaluation of the robustness as a function of spatial frequency preference in the second part is not quite satisfactory. Why averaging the accuracy of the common corruption dataset over the different corruptions after making insightful comparisons between the different severity levels and categories of corruptions in the first part of the paper? It would be good to see the results of Figure 5b separately for the categories and severities. In particular, there seem to be clearly different predictions. In general, it would be most helpful for the reader to perform the same kind of analysis and show the same kind of plots in both parts of the paper which seems possible for almost all analyses in the paper.

3) It would be helpful to include a wider range of baseline models in the second part of the paper (in particular for the mixing experiment and the adversarial perturbations). While it seems plausible that non-robust models prefer higher spatial frequencies compared to robust models, there is only one baseline model presented and it is not clear to what extent its spatial frequency preference might be a function of the specific architecture (which is a WideResNet28-10). Optimally, there would be a baseline model for each of the architectures of the robust models (Table 2). A compromise would be to include a set of baseline models that come close in representing the architectures in Table 2 but do not require model training (i.e., publicly available pretrained models). At the very least one could add a ResNet-18 as additional baseline model.

4) The focus of the paper is on machine learning models. The authors emphasize the importance and value of integrating machine learning and neuroscience which I fully agree with and I commend their approach. However, given that framing, I was missing a treatment (possibly in the discussion section) of the biological system that inspired the computational hypothesis here in a data-driven way. What do we know about whether the biological circuits indeed blur the image? Are psychophysics results and known spatial frequency preferences of mice and primate visual cortices consistent with a blurring hypothesis and the notion that this would increase robustness?

5) Please include more details about the overall procedure (e.g., in an additional procedure subsection in the Methods section) with a level of detail that would allow others to reproduce the analysis. First of two examples: it is unclear what the training and test sets are for adversarial evaluation of the neural regularized models (what is "the full testing dataset"?). Second: “a fixed set of 1000 images were selected from testing set and the incorrect class targets for each image was also fixed". Quite a few details are missing here including what differs between the procedure in the first and the second part of the paper.

Minor comments:

6) It might be very informative to plot the spectra of the blur and pca filters (i.e., of the preprocessing layers) juxtaposed to the corruption spectra (Fig. 7) to understand which part of the robustness results (for the common corruptions) can be explained by a simple filtering explanation.

7) How did the green curve in Figure 5b come about? Please also plot the actual datapoints to which the curve was fit. In addition, it is unclear whether the blur models underlying this curve are all trained with different fixed sigma or whether they correspond to a sweep of sigma on a model trained with a single fixed sigma. See my related point about more detailed description of the procedure.

8) Figure 5b, 6b. Given that the performance of these models on clean images is quite different - wouldn't it make more sense to analyze the decrement in performance due to image corruption (compared to clean image performance) instead of the accuracy on the corrupted images?

9) The second part of the paper should - like the first part - report standard means and standard deviations for the relevant measures.

10) End of last paragraph on p. 3. Reference should be to Fig 2 c, g (not c, f)

11) "Since the mouse regularized models in Fig. 2 are trained on grayscale CIFAR10, it is not included in this comparison" (p. 4). Are there multiple mouse models or just one?

12) The term "corruption accuracy" is a misnomer and slightly confusing when encountering it in Fig. 5b. The legend (in conflict with the y axis label) states "corruption robustness” which is an ambiguous term as well. Accuracy (on corrupted images) would seem like the most accurate term which is also being used in the first part of the paper.

13) "adv" and "crp" are not explained / spelled out in the figure legends or the text.

14) colorbars missing for the power spectra in Fig. 2 and Fig. 4.

15) Even though this is uncommon in machine learning – I would appreciate considerations regarding statistical inference. The standard error of most estimates is a function of the number of evaluation samples and can therefore be made arbitrarily low. Reporting test statistics is therefore somewhat meaningless – but this fact could be made explicit in a short paragraph in the Methods section. It could also be made explicit that the kind of implicit statistical inference made in the paper is at the level of individual models (e.g., the specific instance of a single robustly trained model vs. a specific non-robust model) and not at the level of model classes (e.g., adversarially trained models vs. baseline models).

**Have the authors made all data and (if applicable) computational code underlying the findings in their manuscript fully available?**

Reviewer #1: **No: **The authors claim that the data and the codes will be public later.

Reviewer #2: Yes

Reviewer #3: **No: **The authors state that data and code will be made publicly available in a public repository but I cannot assess to what level of detail and when this will happen.

PLOS authors have the option to publish the peer review history of their article (what does this mean?). If published, this will include your full peer review and any attached files.

Reviewer #1: **Yes: **Chengxu Zhuang

Reviewer #2: No

Reviewer #3: No
---

## [Decision Letter · Decision Letter 1]

14 Nov 2022

Dear Li,

Thank you very much for submitting your manuscript "Robust deep learning object recognition models rely on low frequency information in natural images" for consideration at PLOS Computational Biology.

As with all papers reviewed by the journal, your manuscript was reviewed by members of the editorial board and by several independent reviewers. In light of the reviews (below this email), we would like to invite the resubmission of a significantly-revised version that takes into account the reviewers' comments.

We cannot make any decision about publication until we have seen the revised manuscript and your response to the reviewers' comments. Your revised manuscript is also likely to be sent to reviewers for further evaluation.

Sincerely,

Xuexin Wei

Academic Editor

PLOS Computational Biology

Wolfgang Einhäuser

Section Editor

PLOS Computational Biology

Reviewer's Responses to Questions

**Comments to the Authors:**

Reviewer #1: I want to first thank the authors for their work addressing my concerns. They have added new experiments and results, answered my questions, and provided more explanations and discussions. With these modifications, I now recommend accepting the paper.

However, the newly added ImageNet ResNet-50 results make me feel that the robust networks on small and larger resolutions differ significantly on why they are robust. This low-frequency preference matters more for small resolution compared to large resolution. This resolution-relevant difference may also explain why mouse-regularized networks show a much higher preference for low frequency, as their visual systems have lower acuity. The results of this work are still important, as they uncover what underlies the robustness of small-resolution robust networks and report that the robustness in large-resolution networks may require more mechanisms. Given that the large resolution networks are much more widely used in real-world applications, I encourage the authors to think more about how to explain and further facilitate the robustness of those networks.

No attachment is uploaded.

Reviewer #3: I reviewed the revised version of the manuscript and I appreciate the authors’ responses. However, the authors missed responding to one major and a couple of minor concerns in the rebuttal. I could also not find them addressed in the revised manuscript. I will reiterate them and/or quote from my last review.

Major:

1) I had asked the authors to include more baseline models in their analysis (my third point in the comments to the authors), which was not addressed in the rebuttal. The authors propose a computational principle for robust object recognition. Namely, that low spatial frequency preference is one cause of model robustness. Evidence is presented only for robustly trained models with the exception of a single baseline architecture. Spatial frequency preferences may vary quite substantially between architectures with different spatial integration properties. The correlations between spatial frequency preference and robustness (e.g., Fig 4c and 5b) might therefore become less clear-cut if a wider range of non-robustly trained baseline models is considered. This would restrict the proposed computational mechanism from a general principle to the special case of models trained for robustness. Currently, there is a single baseline model (WideResNet28-10) while the robust models have a wider variety of architectures (WideResNet-70-16, WideResNet-28-10, WideResNet-34-20, ResNeXt29-32x4d, PreActResNet-18, ResNet-18).

It should be quite straightforward to include baseline models of at least a few more of the other architecture types (WideResNet-70-16, WideResNet-28-10, WideResNet-34-20, ResNeXt29-32x4d, PreActResNet-18, ResNet-18) for the CIFAR10 part and I ask the authors to include these analyses in the revised manuscript.

Minor:

2) “9) The second part of the paper should - like the first part - report standard means and standard deviations for the relevant measures.”

3) “11) "Since the mouse regularized models in Fig. 2 are trained on grayscale CIFAR10, it is not included in this comparison" (p. 4). Are there multiple mouse models or just one?” -> It would be helpful for the reader to very briefly clarify in the manuscript or supplement what those models are (why not just one VGG19 mouse model?). Not every reader might be deeply familiar with the details of the respective publication.

4) “13) "adv" and "crp" are not explained / spelled out in the figure legends or the text.”

**Have the authors made all data and (if applicable) computational code underlying the findings in their manuscript fully available?**

Reviewer #1: Yes

Reviewer #3: Yes

PLOS authors have the option to publish the peer review history of their article (what does this mean?). If published, this will include your full peer review and any attached files.

Reviewer #1: No

Reviewer #3: No
---

## [Decision Letter · Decision Letter 2]

6 Feb 2023

Dear Li,

We are pleased to inform you that your manuscript 'Robust deep learning object recognition models rely on low frequency information in natural images' has been provisionally accepted for publication in PLOS Computational Biology.

Best regards,

Xue-Xin Wei

Academic Editor

PLOS Computational Biology

Wolfgang Einhäuser

Section Editor

PLOS Computational Biology

Reviewer's Responses to Questions

**Comments to the Authors:**

Reviewer #3: The authors have addressed all my concerns.

Please make sure to properly cite the URL of the cifar10 model repository (reference [38]) in the paper.

**Have the authors made all data and (if applicable) computational code underlying the findings in their manuscript fully available?**

Reviewer #3: Yes

PLOS authors have the option to publish the peer review history of their article (what does this mean?). If published, this will include your full peer review and any attached files.

Reviewer #3: No

---

## [Editor Report · Acceptance letter]

22 Mar 2023

PCOMPBIOL-D-22-00328R2 

Robust deep learning object recognition models rely on low frequency information in natural images

Dear Dr Li,

I am pleased to inform you that your manuscript has been formally accepted for publication in PLOS Computational Biology. Your manuscript is now with our production department and you will be notified of the publication date in due course.

With kind regards,

Zsofia Freund
